# The Effect of κ-Carrageenan Proportion and Hot Water Extract of the *Pluchea indica* Less Leaf Tea on the Quality and Sensory Properties of Stink Lily (*Amorphophallus muelleri*) Wet Noodles

**DOI:** 10.3390/molecules27165062

**Published:** 2022-08-09

**Authors:** Paini Sri Widyawati, Thomas Indarto Putut Suseno, Anna Ingani Widjajaseputra, Theresia Endang Widoeri Widyastuti, Vincentia Wilhelmina Moeljadi, Sherina Tandiono

**Affiliations:** Food Technology Study Program, Agricultural Technology Faculty, Widya Mandala Surabaya Catholic University, Dinoyo Street Number 42-44, Surabaya 60265, Indonesia

**Keywords:** *Amorphophallus muelleri*, *Pluchea indica* Less, wet noodles, quality and sensory properties

## Abstract

The study aims to determine the effect of the proportion of κ-carrageenan and the hot water extract of pluchea leaf tea on the quality and sensory properties of stink lily wet noodles. The research design is a randomized block design with two factors, i.e., the difference in the proportion of κ-carrageenan (K) (0, 1, 2, and 3% *w*/*w*) and the addition of the hot water extract of the *Pluchea indica* Less leaf tea (L) (0, 15, and 30% *w*/*v*), with 12 treatment levels (K0L0, K0L1, K0L2, K1L0, K1L1, K1L2, K2L0, K2L1, K2L2, K3L0, K3L1, K3L2). The data are analyzed by the ANOVA at *p* < 5% and continued with the Duncan’s multiple range test at *p* < 5%, and the best treatment was determined by the spider web method based on sensory assay by a hedonic method. The proportions of κ-carrageenan and the concentration of pluchea tea extract had a significant effect on the cooking quality and sensory properties. However, the interaction of the two factors affected the swelling index, yellowness (b*), chroma (C), hue (h), total phenol content (TPC), total flavonoid content (TFC), and DPPH free radical scavenging assay (DPPH). The best treatment of wet noodles was K2L0, with a preference score of 15.8. The binding of κ-carrageenan and phenolic compounds to make a networking structure by intra- and inter-disulfide bind between glucomannan and gluten was thought to affect the cooking quality, sensory properties, bioactive compounds (TPC and TFC), and DPPH.

## 1. Introduction

Noodles are a rice-substitute commodity that are much favored by the public [1], especially in China, Indonesia, India, Japan, Vietnam, and the United States. Basically, noodles are divided into wet noodles and dry noodles. Indonesia is the country that consumes the second-highest amount of noodles in the world [2]. In 2017, the consumption of instant noodles became 180.2 packets per head in the world, and in Indonesia, the consumption of wet noodles shows that all age groups and education levels enjoy them [3]. Meanwhile, instant noodle consumption in 2020 reached 12,640 million portions [2]. Noodles are generally made from wheat flour; however, the use of local food ingredients based on carbohydrates, including stink lily flour, can reduce the consumption of wheat flour, which is quite high in Indonesia, with the average consumption of wheat flour for the Indonesian population in 2019 being 17.8 kg/capita/year [4].

Stink lily flour (*Amorphophallus muelleri*) is a group of *Araceae* that contains glucomannan oligosaccharides of around 15–64% dry base [5], or more than 60% [6]. Glucomannan is a heteropolysaccharide consisting of 67% D-mannose and 33% D-glucose, and has β-1,4 and β-1,6 glycoside bonds [7] which can reduce body weight, blood sugar content levels, LDL cholesterol levels, and prolong gastric emptying time [8,9]. Stink lily flour has a glycemic index of 85, which is lower than the glycemic index of glucose (which is considered to be 100 [7]). The use of stink lily flour in the manufacture of wet noodles is able to replace the role of the gliadin and glutenin proteins in the formation of gluten with an elastic texture [10]. Polysaccharides in stink lily flour can dissolve in water to form a thick solution, form a gel, expand, melt like agar [11], and can increase the elasticity and cohesiveness [12,13] with increasing α-helix and β-sheet structures of wet noodles [13]. However, the higher the substitution of stink lily flour, the lower the texture preference, because the noodles break easily and are sticky [12]. Therefore, it is necessary to add hydrocolloids, including carrageenan, because they can increase elasticity [14]. The combination of the use of glucomannan and carrageenan can form a strong and elastic gel [15].

The addition of the hot water extract from pluchea leaf tea in making stink lily wet noodles is expected to increase the functional value of wet noodle products. Pluchea leaves contain nutrients, such as: protein 1.79 g/100 g, fat 0.49 g/100 g, ash 0.20 g/100 g, insoluble fiber 0.89 g/100 g, soluble fiber 0.45 g/100 g, total fiber 1.34 g/100 g, carbohydrates 8.65 g/100 g, calcium 251 g/100 g, β-carotene 1.225 mg/100 g, and vitamin C 30.17 mg/100 g, as well as bioactive compounds, such as: phenolic acid 28.48 ± 0.67 mg/100 g wb (chlorogenic acid 20 ± 0.24 mg/100 g wb, caffeic acid 8.65 ± 0.46 mg/100 g wb), total flavonoids 6.39 mg/100 g wb (quercetin 5.21 ± 0.26 mg/100 g wb, kaempferol 0.28 ± 0.02 mg/100 g wb, myricetin 0.09 ± 0.03 mg/100 g wb), total anthocyanins 0.27 ± 0.01 mg/100 g wb, β-carotene 1.70 ± 0.05 mg/100 g wb, and total carotenoids 8.7 ± 0.34 mg/100 g wb [16], 3-O-caffeoylquinic acid, 4-O-caffeoylquinic acid, 5-O-caffeoylquinic acid, 3,4-O-dicaffeoylquinic acid, 3,5-O-dicaffeoylquinic acid, and 4,5-O-dicaffeoylquinic acid [17,18,19]. Meanwhile, the hot water extract of 2% pluchea leaf tea (2 g/100 mL) contains a total phenolic content of 9.3 mg EAG/g, total flavonoid content 22.0 mg EC/g, DPPH free radical scavenging activity DPPH 27.2 mg EAG/g, and reduced iron ion 10.2 mg EAG/g [20], due to the presence of phytochemicals (alkaloids, flavonoids, phenolics, sterols, cardiac glycosides, phenol hydroquinone, tannins, terpenoids, and saponins) [20], which has been shown to have potential as antioxidants [20,21] and antidiabetics [22]. The effect of using κ-carrageenan and the water extract of pluchea leaf tea on the quality and sensory properties of stink lily wet noodles has not been studied in detail. Therefore, the purpose of this study is to determine the effect of the proportion of κ-carrageenan and the hot water extract of pluchea leaf tea on the quality and sensory properties of stink lily wet noodles.

## 2. Materials and Methods

### 2.1. Reagents and Materials

The compounds 2,2-diphenyl-1-picrylhydrazyl (DPPH), sodium carbonate, gallic acid, and (+)-catechin were purchased from Sigma-Aldrich (St. Louis, MO, USA). Methanol, Folin–Ciocalteu’s phenol, sodium nitric, aluminum chloride, κ-carrageenan, and sodium hydroxide were purchased from Merck (Kenilworth, NJ, USA).

The pluchea leaves as a raw material for making the pluchea leaf tea were collected from gardens around the city of Surabaya. The specification of the pluchea plant was according to the GBIF taxon ID number database: 3132728. The stink lily flour was obtained from the stink lily flour processing industry in East Java. The specification of the ttink lily plant was according to the GBIF taxon ID number database: 735493731. The wheat flour used was a high-protein flour obtained from the wheat flour processing industry in Indonesia.

### 2.2. Preparation of the Pluchea Leaf Tea

The pluchea leaves on each branch (number 1–6 from) the shoot was collected, sorted, and dried at an ambient temperature for 7 days, until the moisture content was 11.16 ± 0.09% dry base. Then, the dried leaves were powdered to achieve a 45 mesh size [23]. Furthermore, the leaf powder was heated by a drying oven (Binder, Merck KGaA, Darmstadt, Germany) at 120 °C for 10 min. Then, the dried powder of the pluchea leaves was packed as 2 g per tea bag, which was then called pluchea leaf tea.

### 2.3. Preparation of the Hot Water Extract of Pluchea Leaf Tea

The pluchea leaf tea in a tea bag was extracted by hot water at 95 °C for 1 min to achieve 15 and 30% (*b*/*v*) concentrations (Table 1). Then, each concentration of the extract was used to make stink lily wet noodles.

### 2.4. Stink Lily Wet Noodles Making

The stink lily wet noodles were made with a mixture of wheat and stink lily flour and κ-carrageenan at 1, 2, and 3% (*b*/*b*) concentrations. Then, the mixture was added to egg, salt, baking powder, and the hot water extract of the pluchea leaf tea, and kneaded to form a dough by a mixer machine. The dough was then passed through a roller to make face bands of the desired thickness, and was cut through rollers using cutting blades. The formula of the stink lily wet noodle is showed in Table 2.

### 2.5. Stink Lily Wet Noodles Extraction

125 g of each sample of the stink lily wet noodles was weighed (Ohaus, Ohaus Instruments (Shanghai) Co., Ltd., Shanghai, China), and then they were dried by cabinet drying at 60 °C for 4 h to get dried noodles. Next, each sample was powdered by a chopper machine at a second speed for 35 s, and then 20 g of each powdered sample was added to 50 mL methanol by a shaking water bath at 35 °C, 70 rpm, for 1 h. The filtrate was separated by Whatman filter paper grade 40, and the residue was extracted again with same pattern method. The filtrate was collected and dried by a rotary evaporator (Buchi Rotary Evaporator; Buchi Shanghai Ltd., RRT, Shanghai, China) at 0.2–0.3 atm, 50 °C, for 60 min, until a 2 mL extract was achieved. Then, the extract was kept at 0 °C before further study.

### 2.6. Swelling Index Assay

Swelling index, or water absorption, is the ability of noodles to absorb water after gelatinization during the boiling process [24]. The principle of water absorption testing is to determine the amount of water absorbed in wet noodles at a certain temperature and time. The amount of water absorbed in wet noodles can be determined from the difference between the weight of the noodles after and before being boiled divided by the weight of the noodles before boiling [25].

### 2.7. Cooking Loss Assay

Cooking loss is one of the important quality parameters in wet noodles to determine the quality of wet noodles after cooking [26]. The cooking loss test for stink lily wet noodles was carried out to determine the number of solids that came out of the noodle strands during the cooking process, namely, the release of a small portion of starch from the noodle strands.

### 2.8. Determination of the Tensile Strength of Wet Noodles

The tensile strength (elongation) is one important parameter of texture analysis in noodle products. The texture was determined using a TA-XT2 texture analyzer (Stable Micro System Co., Ltd., Surrey, UK), fitted with a 5 kg load cell equipped with the Texture Exponent 32 software V.4.0.5.0 (SMS). The principle of the texture analyzer was to prepare a suitable probe for the test, then place the noodle samples on the table under the probe. The elongations of the noodles were individually tested by putting one end into the lower roller arm slot and sufficiently winding the loosened arm to fasten the noodle end. The same procedure was performed to tighten the other end of the strip to the upper roller arm. Elongation, which was the maximum force to deform and break the noodles by extension, was measured using a test speed of 3.0 mm/s, with a 100 mm distance between the two rollers. Deformations were recorded using the software during the extension, and are expressed as a graph. The elongation at breaking was calculated per gram.

### 2.9. Color Measurement

The noodle samples were measured by a colorimeter (Minolta CM-3500D; Minolta Co. Ltd., Osaka, Japan), and the CIE-Lab L*, a*, and b* values were recorded as described by Rathorel et al. [27]. Then, the L*, a*, and b* values were collected. The L* value was stated as the position on the white/black axis, the a* value the position on the red/green axis, and the b* value the position on the yellow/blue axis. The measurements were carried out in triplicate, and the readings were averaged.

### 2.10. Total Phenol Content Assay

The total phenol content (TPC) of the stink lily wet noodles was analyzed by the spectrophotometric method using the Folin–Ciocateu’s phenol reagent [28]. Principles assay of the TPC assay are the interactions between phenolic compounds and phosphomolybdic/phosphotungstic acid complexes, based on the transfer of electrons in an alkaline medium from the phenolic compounds to form a blue chromophore constituted by a phosphotungstic/phosphomolybdenum complex. The reduced Folin–Ciocalteu’s phenol reagent was detected by a spectrophotometer (Spectrophotometer UV-Vis 1800, Shimadzu, Japan) at λ 760 nm and gallic acid was used as the reference standard compound, and the results are expressed as gallic acid equivalents (mg/kg wet noodles).

### 2.11. Total Flavonoid Content Assay

The total flavonoid content of the samples was measured by the spectrophotometric method, with a reaction between AlCl_3_ and NaNO_2_ with an aromatic ring of flavonoid compounds [29]. Then, the mixture was added to aluminum chloride, resulting in a yellow solution. Next, the addition of the NaOH solution in the mixture caused a red solution, that was measured by a spectrophotometer (Spectrophotometer UV-Vis 1800, Shimadzu, Japan) at λ 510 nm. A standard reference compound of (+) catechin was used, and the results were expressed as catechin equivalents (mg/kg wet noodles).

### 2.12. DPPH Free Radical Scavenging Activity Assay

The DPPH free radical scavenging activity was measured by the spectrophotometric method [30]. This method is used to determine the antioxidant capacity of a compound from an extract or other biological sources, based on the transferring from the odd electron of a nitrogen atom in DPPH being reduced by receiving a hydrogen atom from antioxidants, to result in DPPH-H with a yellow-colored solution. The reaction between the DPPH in methanol solution with the samples was measured by a spectrophotometer (Spectrophotometer UV-Vis 1800, Shimadzu, Japan) at λ 517 nm. The standard reference compound used was gallic acid, and the results were expressed as gallic acid equivalents (mg/kg wet noodles).

### 2.13. Sensory Evaluation

Sensory assay was carried out to determine the level of panelist acceptance of wet noodles substituted with stink lily flour with the addition of carrageenan and the hot water extract of pluchea leaf tea [31]. The test was carried out using the hedonic scale scoring method. This method is designed to measure the level of panelist preference for the product by rating the level of preference for the product being tested. Samples were served in dishes coded with random three-digit numbers that allowed for a completely-randomized design (CRD) trial to be carried out, using 100 untrained panelists with an age range of 17 to 25 years. The hedonic method used in this study was the hedonic scoring method, and the panelists were asked to give a preference score for each sample. The hedonic score used was a value of 1–15, given by the panelists according to the level of preference for the product. Values 0–3.0 indicated “strongly dislike”, values 3.1–6.0 indicated “dislike”, values 6.1–9.0 indicated “neutral”, values 9.1–12.0 indicated “like”, and a value of 12.1–15.0 indicated “very much like”.

Each panelist was faced with 12 (twelve) samples and a questionnaire containing test instructions, and was asked to give each sample a score according to their level of preference. The parameters tested were taste, aroma, texture, color, and overall acceptance of wet noodles substituted with stink lily flour. The best treatment of the samples was determined by the spider web method, that was correlated by the large area of graph [32].

### 2.14. Experiment Design and Statistical Analysis

The research design of the physicochemical assay used was a randomized block design with two factors, i.e., differences in the proportion of the κ-carrageenan (K) and differences in the concentration of the pluchea tea extract (L) added to the wet noodles. The proportion of κ-carrageenan consisted of four treatment levels, including 0% (K0), 1% (K1), 2% (K2), and 3% (K3), and the concentration of pluchea tea extract consisted of three levels, i.e., 0% (L0), 15% (L1), and 30% (L2). Each treatment was repeated three times in order to obtain 36 experiment units. The sensory test used a completely randomized design (CRD) on 100 untrained panelists.

The data, before further analysis, was determined by the normal distribution and homogeneity tests. Then, the data were presented as the mean ± SD of the triplicate determinations, and were analyzed using ANOVA at *p* < 5%. If the results of the ANOVA test had a significant effect, then the DMRT (Duncan Multiple Range Test) was proceed with at *p* < 5% to determine the level of treatment that gave significantly different results. The analysis used SPSS 17.0 software (SPSS Inc., Chicago, IL, USA).

## 3. Results

### 3.1. Cooking Quality

The evaluated cooking properties of the stink lily wet noodles were shown in Table 3, and the stink lily wet noodles product was shown in Figure 1. The level of cooking was estimated by the moisture content, swelling index, cooking loss, and tensile strength from the noodles. Based on the statistical analysis by ANOVA at *p* < 5%, the increasing κ-carrageenan proportion was shown to have created a significant difference of moisture content of the wet noodles, but the addition of the pluchea leaf hot water extract, and the interaction effect of the proportion of κ-carrageenan and the addition of the extract, had no influence on the moisture content of the wet noodles. The moisture content value ranged from 62.83 ± 0.58 %wb to 65.83 ± 0.22 %wb. The sample K3L0 had the highest moisture content, and K0L2 had the lowest moisture content. Furthermore, the addition of the κ-carrageenan proportion or the pluchea tea extract had significantly different effects, such as the interaction effect of the addition of the two parameters, on the swelling index or water absorption value of the wet noodles, based on the statistical analysis at *p* < 5%. The water absorption value ranged from 142.25 ± 0.39% to 162.21 ± 0.25%. The treatment with the lowest swelling index was K0L2, and the highest was K3L0. Contrary to this, the cooking loss of the wet noodles decreased significantly with the addition of the proportion of κ–carrageenan, but increased significantly with the addition of the pluchea tea extract. The cooking loss of the wet noodles ranged 17.83 ± 0.4% to 20.13 ± 0.7%. K0L2 was the treatment with the biggest cooking loss, and K3L0 was the treatment with the smallest cooking loss. The tensile strength value of the stink lily noodles was significant different because there was an interaction effect of the κ-carrageenan proportion and the pluchea tea extract addition. The tensile strengths of the wet noodles ranged 0.096 ± 0.004 N to 0.174 ± 0.015 N. The analysis of the stink lily noodles color showed that lightness had a significant increase with an increasing proportion of carrageenan and decreased with the increase in the amount of pluchea tea extract used, because of the color effect of the carrageenan and pluchea tea extracts. The lightness of the wet noodles ranged from 67.80 ± 0.22 to 74.50 ± 0.23. The effect of the κ-carrageenan and the pluchea tea extract color also significantly influenced the redness of wet noodles. The redness of the wet noodles ranged from 1.20 ± 0.04 to 3.30 ± 0.23. The interaction effect of the κ-carrageenan and the pluchea tea extract addition appeared on the yellowness, chroma, and hue values. The yellowness, chroma, and hue values ranged 16.90 ± 0.27 to 30.00 ± 0.07, 17.00 ± 0.28 to 30.10 ± 0.03, and 83.70 ± 0.07 to 86.40 ± 0.02, respectively.

### 3.2. Bioactive Compounds and DPPH Free Radical Scavenging Activity (DPPH)

The bioactive compounds analyzed include the total phenol content (TPC) and the total flavonoid content (TFC). The analysis data shows that the TPC and TFC increased significantly due to the interaction effect between the proportion of κ-carrageenan and the addition of the pluchea tea extract (Table 4). The results of the analysis show that the control sample K0L0 had the lowest total phenol, i.e., 0.341 ± 0.034 mg GAE/kg of dry noodles. The K2L2 sample had the highest total phenol, which was 1.196 ± 0.027 mg GAE/kg of dry noodles. This result is suitable with the TFC value of wet noodles, where K0L0 had the lowest TFC value and K2L2 had the highest value, i.e., 0.057 ± 0.004 and 1.336 ± 0.046 mg CE/kg of dry noodles, respectively. The high and low values of TPC and TFC were correlated with AOA. The higher the TPC and TFC values, the higher the DPPH value. The DPPH free radicals scavenging activity of the wet noodles was determined to be significant by the interaction effect of adding the proportion of κ-carrageenan and the pluchea tea extract. K0L0 had the lowest DPPH value and K2L2 had the highest DPPH value, i.e., 0.414 ± 0.006 and 0.758 ± 0.009 mg GAE/kg of dry noodles, respectively.

### 3.3. Sensory Properties

The evaluated sensory properties of the stink lily noodles are shown in Table 5. The sensory parameters that were analyzed included aroma, color, taste, texture, and overall acceptability. The method used for the sensory assay of the stink lily noodles was a hedonic scale scoring, or a test of the level of consumer preference for a product by giving an assessment or score on a certain trait [33]. The organoleptic testing of the stink lily wet noodles was presented to 100 untrained panelists aged 17–25 years. The panelists were asked to give scores or numbers based on their level of preference for certain treatments. The value score used was 1–15, where a value of 0–3.0 indicated “strongly dislike”, a value of 3.1–6.0 indicated “does not like”, a value of 6.1–9.0 indicated “neutral”, a value of 9.1–12.0 indicated “like”, and a value of 12.1–15.0 indicated “like very much”. The results of the statistical tests by ANOVA at *p* < 5% show that the interaction effect of each treatment significantly influenced the panelists’ preference for noodle color. The preference value of the stink lily noodles’ color was ranged from 9.12 to 12.02 (like). The highest color preference value was the control treatment (K0L0), and the treatment with the lowest color preference value was K0L1. The interaction effect of each treatment had a significant impact on the panelists’ preference for the aroma. The preference value for the aroma of the wet noodles was ranged from 8.29 to 11.58 (neutral–like). The treatment with the highest preference value was the stink lily noodles K1L0, and the treatment with the lowest preference value was the K3L2 treatment. The preference value of noodles taste was only affected by the κ-carrageenan proportion or the extract concentration. The preference value for the wet noodle taste ranged from 8.18 to 11.08 (neutral–like). The treatment with the highest taste preference value was K2L0, while the treatment with the lowest taste preference value was K0L2. Increasing the proportion of carrageenan to 2% increased the preference value for taste, however after that it decreased with the addition of the proportion of 3% carrageenan. Meanwhile, increasing the concentration of the pluchea leaf tea extract decreased the panelists’ preference for taste. The results of the statistical tests using ANOVA at *p* < 5% show that the addition of the extract only significantly influenced the panelists’ preference for the noodles texture. In this study, the preference for the wet noodles texture ranged from 9.46 to 11.66 (like).

Increasing the addition of the extracts decreased the level of texture preference of the wet noodles. The overall preference value of the wet noodles shows that there is an interaction effect between the proportion of κ-carrageenan and the addition of the pluchea tea extract. The overall preference value of the wet noodles ranged from 8.62 to 11.24 (neutral–like). The treatment with the highest preference value was stink lily noodles K0L0, and the treatment with the lowest preference value was K3L2. The determination of the best treatment for the differences in the proportions of κ-carrageenan and the addition of the pluchea tea extract on the wet noodles was determined using the spider web method, based on organoleptic parameters (color, aroma, taste, texture, and overall). The spider web graph can be seen in Figure 2. The data shows that the treatment with the largest area was K2L0, i.e., wet noodles with the proportion of κ-carrageenan 2% and with the addition of 0% pluchea tea extract. The area of the K2L0 treatment area was 79.16 cm^2^, and had a preference score of 15.8 (very like).

## 4. Discussion

### 4.1. Cooking Quality

Moisture content is a major parameter of the stinky silky wet noodles that shows the amount of water contained in food product that determines rheological characteristics, the chemical, physical, and sensory properties, and the shelf life of the food product [34]. The result show that the κ-carrageenan addition gave a significant difference of moisture content to the wet noodles. The κ-carrageenan is a hydrocolloid that has a group of sulfate- and water-soluble polysaccharides [35,36,37], and composes an ester sulfate content of about 25–30% and a 3,6-anhydro-galactose (3,6-AG) of about 28 to 35% [36,38]. This anionic carrageenan can interact very tightly with water molecules [39], and can collaborate with the glucomannan from the stink lily on the gelation process [40] by making intra- and inter-disulfide binds at the network structure of gluten [13,41]. The κ-carrageenan can bind with limited free water molecules, form complex compounds with water, and interact with gluten networks [13]. However, the mobility of water mainly depends on changes in the hydrogen bond structure. The presence of hydrophilic components such as proteins, carbohydrates, glucomannan, κ-carrageenan, and polyphenolic compounds in the wet noodles can be involved in hydrogen bonding with the water molecules that determine water mobility [13,34,41,42]. Thus, increasing the proportion of k-carrageenan used in the manufacture of stink lily noodles can increase the amount of free water and weakly bound water in the wet noodles. The interaction between the hydroxyl functional groups in carrageenan and water molecules supports an increase in the water content of wet noodles.

The swelling index, or water absorption, is the ability of a product to absorb water, which is influenced by particle size, chemical composition, and water content [43]. The research shows that the interaction of the κ-carrageenan proportion and the hot water extract of the pluchea leaf tea addition had a significant impact on the swelling index properties of wet noodles. The swelling index of the wet noodles was determined by the presence of κ-carrageenan, protein, starch, glucomannan, and bioactive compounds of the pluchea tea extract in the dough. Li et al. [13] stated that the glutelin proteins of wheat flour can allow intra- and inter-molecular disulfide bonds to create a fibrous shape, and that then the globular gliadin protein of the wheat flour can be bound at the glutenin skeleton by non-covalent bonds to be a unique networks structure of gluten. The addition of the glucomannan of the stinky silky flour as a non-ionic hydrocolloid has good water holding capacity, and can be made into a stronger three-dimensional network structure. Chen et al. [44] stated that the glucomannan can fill the number of holes of the network structure of gluten that make a structure dense and stable. Li et al. [13] underlined that glucomannan has many hydroxyl groups in the structure that can be bound tightly with water by electrostatic forces and hydrogen bonds. Huang et al. [41] stated that the presence of κ-carrageenan in dough can be synergist with glucomannan to change sulfhydryl groups into disulfide bonds in protein. Widyawati et al. [20] stated that the bioactive compounds of pluchea tea extract are alkaloids, flavonoids, phenolics, phenol hydroquinone, saponins, tannins, sterols, terpenoids, and cardiac glycosides. Meanwhile, Suriyaphan [16] noted that pluchea leaves contain 1.79 g/100 g protein, 0.49 g/100 g fat, 0.20 g/100 g ash, 0.89 g/100 g insoluble fiber, 0.45 g/100 g dissolved fiber, total fiber 1.34 g/100 g, carbohydrates 8.65 g/100 g, calcium 251 g/100 g, β-carotene 1.225 g/100 g, and vitamin C 30.17 g/100 g, as well as phenolic acid bioactive compounds 28.48± 0.67 mg/100 g body weight (chlorogenic acid 20 ± 0.24 mg/100 g body weight, caffeic acid 8.65 ± 0.46 mg/100 g body weight), total flavonoids 6.39 mg/100 g body weight (quercetin 5.21± 0.26 mg/100 g body weight, kaempferol 0.28 ± 0.02 mg/100 g body weight, myricetin 0.09 ± 0.03 mg/100 g body weight), total anthocyanins 0.27 ± mg/100 g body weight, β-carotene 1.70 ± 0.05 mg/100 g body weight, and total carotenoids 8.7 ± 0.34 mg/100 g body weight. Vongsak et al. [17], Ruan et al. [18], and Chan et al. [19] proved that pluchea leaves contain 3-O-caffeoylquinic acid, 4-O-caffeoylquinic acid, 5-O-caffeoylquinic acid, 3,4-O-dicaffeoylquinic acid, 3,5-O-dicaffeoylquinic acid, and 4,5-O-dicaffeoylquinic acid. Schefer at al. [45] noted that phenolic acid can be bound with proteins and carbohydrates by non-covalent interactions, i.e., hydrophobic interaction, hydrogen bonding, electrostatic interaction, Van der Waals interaction, and π-π stacking. The presence of the κ-carrageenan proportion and the pluchea tea extract addition that differed caused a change of composition and various interactions of compounds that determined different swelling indexes of the wet noodles.

The cooking loss of the stink lily noodles was influenced significantly by the κ-carrageenan proportion and the extract addition, but interaction of two factors had an insignificant difference. The phenomena were caused by the κ-carrageenan and bioactive compounds of the pluchea tea extract, especially phenolic compounds, that involved the interaction with proteins and carbohydrates in dough. κ-carrageenan can stabilize and supported a rigid structure of gluten. The hydrocolloid can avoid the starch gelatinization process because it can bind tightly with water molecules, causing lower water activity. According to Li et al. [13], Herawati [14], and Huang et al. [41], κ-carrageenan can trap the free water molecules that starch can’t absorb, and require higher energy to break the energy barrier required for the starch gelatinization process. However, Zhu [42], Amoako and Awika [46], and Schefer et al. [45] clarified that starch can be bound with polyphenol, including hydrophobic and electrostatic interactions and hydrogen bonds, and that the hydrogen bond is dominant binding forces. This interaction can support the releasing of the amylose of the starch gelatinization process that the cooking loss increased at the higher pluchea tea extract addition. The presence of the polyphenol compounds of the pluchea tea extract causes water competition with the glutenin and gliadin of wheat flour that inhibited interaction between glutenin and gliadin to form gluten. According to Amoako and Awika [46], gluten and gliadin in a random coil structure can easily be aggregated by phenolic compounds and starch when undergoing a gelatinization process where amylose interacts with polyphenolic compounds through hydrogen bonds and hydrophobic interactions. More protein in the form of random coil structures causes the protein to easily interact with polyphenols and come out of the noodles during the cooking process, meaning that the cooking loss increases.

The tensile strength of the stink lily was influenced significantly by the κ-carrageenan proportion and the extract addition. The increasing κ-carrageenan proportion increased the tensile strength because this hydrocolloid could be made strong by cross linking through the inter-molecular and intra-molecular bonds involving the glutenin and gliadin of wheat flour protein and the glucomannan of stink lily flour. The more networks that are formed between the components of the noodles, the greater the effect on the tensile strength of wet noodles, and vice versa [13,47]. The synergism effect of the wet noodles component determined the water bind capacity and water mobility, that in turn established texture properties of wet noodles; this statement is supported by Li et al. [34], Saha and Bhattacharya [40], and Huang et al. [41]. Diniyah et al. [48] also reported that the addition of hydrocolloids in the noodles-making process increases their viscosity and water absorption, due to the water binding and holding properties of hydrocolloids that can form gel. However, the addition of the pluchea tea extract caused the tensile strength to decline, because the polyphenol compounds of the pluchea tea extract induced a breakdown of the networking structure among the components of the dough because there was water competition among them. Furthermore, the polyphenol compounds could react with starch and protein because the formation of the gluten network was disrupted, and also gliadin and glutenin in the form of random coils and starch could have undergone an excessive gelatinization process. This opinion is supported by Li et al. [13], Zhu [42], and Huang et al. [41]. κ-carrageenan has a yellowish white color, and has the ability to bind water molecules so that it increases the lightness of wet noodles, while the pluchea tea extract contains polyphenolic compounds, such as tannins, which can give the noodles a brown color so that the lightness level is reduced. This opinion is supported by Widyawati et al. [20] and Necas et al. [38]. The increase in yellowness was in line with the increase in lightness, because the higher water content value was caused by the ability of κ-carrageenan to bind water molecules, thereby increasing the brightness; meanwhile. the brown color contribution of the pluchea tea extract gave a brownish–yellow color to the wet noodles, the intensity of which increased as indicated by the increased chroma value.

### 4.2. Bioactive Compounds and the DPPH Free Radical Scavenging Activity

The stinky silky wet noodles K0L0 had the lowest TPC and TFC because there was a contribution of phenolic content from the wheat flour and egg. Punia et al. [49] said that wheat flour had phenolic acids including ferulic, caffeic, and p-coumaric acid. Moreover, the presence of TFC in the K0L0 sample is thought to be due to the presence of a thiol group in egg white, which is able to chelate metal ions and is able to be conjugated with saccharides [50], as well as the 3,5-diacetyltambulin compounds from stink lily flour [51]. Meanwhile, the TFC and TPC values in the K2L2 sample were dominantly contributed to by the presence of phytochemical compounds in the pluchea tea extract. Widyawati et al. [20] explained that there are phytochemical compounds in the pluchea tea extract. Suriyaphan [16], Vongsak et al. [17], Ruan et al. [18], and Chan et al. [19] also emphasized that pluchea leaves contain phenolic acids and flavonoids. The existence of a non-significant difference between treatments in the TPC and TFC assays indicated an interaction between the components in the dough, and that it affected the presence of free hydroxyl groups that could bind to the Folin–Ciocalteu’s phenol reagent. As described by Li et al. [13], Huang et al. [41], Zhu [42], Schefer et al. [45], and Amoako and Awika [46], glutenin, gliadin, glucomannan, κ-carrageenan, and polyphenol compounds are involved in the formation of network structures in the dough so as to determine the quality of the wet noodles. The interactions that occur involve various non-covalent interaction mechanisms that affect the presence of free hydroxyl groups. The TPC and TFC values of wet noodles in each treatment affected the DPPH free radical scavenging activity (DPPH). They determined the DPPH of wet noodles, and were usually positively correlated. Niroula et al. [52] said that TPC and DPPH were strongly correlated in seeds, sprouts, and grasses of corn (*Zea mays* L.). Lim et al. [53] also stated that there is an excellent correlation coefficient between the TPC, TFC, and antioxidant activities of the *Phaleria macrocarpa* fruit. Adebiyi et al. [54] explained that the high level of flavonoids and phenols in plants caused the antioxidant activity of the *Grewia carpinifolia* extract. The antioxidant activity of phenolics is related to their redox properties, which induced them to act as reducing agents, hydrogen donors, singlet oxygen quenchers, and metal chelators. Rahman et al. [55] underlined that the DPPH free radical scavenging activity of the polyphenol compounds of the *T. pallida* extract was determined by the hydrogen donating ability, with which it highly correlated. The potency of wet noodles as AOA was determined by the reduced capability of the DPPH free radical solution color from purple to yellow color.

### 4.3. Sensory Properties

The analysis of the sensory properties of the stink lily noodles was conducted by the hedonic scale scoring method, with attributes including the preferences of color, taste, aroma, texture, and overall. The result of the color preference test shows that the control treatment (K0L0) was the highest value, because the treatment without the addition of pluchea tea extract did not change the color of the wet noodles, meaning that they remained a yellowish–white. Then, the treatment with the lowest color preference value was K0L1, due to the addition of this extract decreasing the panelists’ preference for the color because the noodles were darker and a brownish color. Suriyaphan [16] and Widyawati et al. [20] said that the color of the pluchea tea extract contributed to the changing of the color of this wet noodle due to its tannins, flavonoids, and chlorophyll. However, in this study, increasing the extract concentration did not significantly affect the panelists’ preference for color when the proportion of κ-carrageenan increased, because the addition of the stink lily flour and the κ-carrageenan would increase the lightness of the wet noodles, so that the produced color of the wet noodles was brighter and preferred by panelists. This data is supported by the data from color rider analysis, where the results of the sensory test by the panelists are in line with the decrease in the lightness value, the increase in the reddish and yellowish values, the hue value that showed the yellow–red color, and the chroma value that showed an increase in color intensity. The panelist’s preference of the aroma from the wet noodles was determined to be due to the aroma from the material used to make the wet noodles or the interaction of the aroma produced from the reaction among the material composition. According to Ramdani et al. [56], stink lily flour has a musty aroma, and all of the wet noodles produced had a musty smell. Meanwhile, according to Fitantri et al. [57], κ-carrageenan is unscented, meaning that it has no aroma to contribute to the wet noodles. The addition of the pluchea tea extract decreased the panelists’ preference for the aroma of the wet noodles because the addition of the extract caused the wet noodles to smell like leaves (floral), which was unpleasant and so the panelists did not like it. Fragrant or unpleasant aromas come from volatile compounds contained in the pluchea leaves. According to Widyawati et al. [58], pluchea leaves have 66 volatile compounds, and these volatile compounds play a role in forming the aroma in the hot water extract of the pluchea leaf tea. According to Lee et al. [59], pluchea leaves contain volatile compounds contributed to by aliphatic aldehyde group compounds, or aromatic compounds, that give a distinctive aroma, therefore the presence of these compounds in steeping water can give a specific aroma, i.e., fragrant (floral) in wet noodles. There was a difference in the effect of the proportion of the κ-carrageenan and pluchea leaf tea extract on taste due to the contribution of taste produced by the carrageenan and the extract. According to Ramdani et al. [56] and Haryu et al. [60], stink lily flour and κ-carrageenan do not have a distinctive or neutral taste, so increasing the concentration of κ-carrageenan gives a higher preference value because noodles are considered to have a better texture that contributes to the assessment of taste. The increase in the concentration of pluchea tea extract caused the taste preference value of the noodles to decrease significantly, which was due to the presence of tannins, catechins, and phenolic compounds in the pluchea tea extract, which were determined to be bitter and slightly astringent. The effect of the κ-carrageenan proportion and tea extract to make wet noodles influenced the panelist’s preference of texture because this hydrocolloid can cause strong cross linking through inter-molecular and intra-molecular bonds involving the glutenin and gliadin of wheat flour protein and the glucomannan of stink lily flour, which determined the water bind capacity and water mobility [34,40,41]. The presence of the polyphenol compounds of the pluchea tea extract can cause a breakdown of the networking structure among the components of the dough because of water competition of them. Li et al. [13], Huang et al. [41] and Zhu [42] reported that the polyphenol compounds could react with starch and protein to disrupt the gluten network and cause starch to undergo an excessive gelatinization process. The difference in the proportion of the κ-carrageenan changes the overall preference value of wet noodles to be significantly different overall compared to the control, because the proportion of the κ-carrageenan influenced the all-sensory attribute (color, aroma, and taste). The addition of the pluchea tea extract in the wet noodles decreased the overall preference value because the addition of the extract affected the organoleptic characters tested due to the content of secondary metabolites of the pluchea leaves, such as flavonoids, phenols, and tannins that could affect the taste, aroma, color, and texture of the noodles. The spider web graph showed that K2L0 was the best treatment of the stink lily wet noodles. This was also supported by it containing better physicochemical properties than the control, including yellowish–white wet noodles, better swelling index, lower cooking loss, higher tensile strength value, and lower moisture content. However, this K2L0 treatment did not have the highest TPC, TFC, and AOA, i.e., 0.500 ± 0.045; 0.089 ± 0.008; and 0.532 ± 0.005, respectively.

## 5. Conclusions

The use of κ-carrageenan proportions and pluchea leaf tea extract have a significant effect on the cooking quality and sensory properties of stink lily wet noodles. Statistical analysis at *p* < 5% showed that there was an interaction effect of the proportion of the κ-carrageenan and the pluchea leaf tea extract on the swelling index, yellowness, chroma, hue, TPC, TFC, DPPH, the preference value for color, aroma, and overall acceptance. While the moisture content of the wet noodles was only affected by the proportion of κ-carrageenan, the tensile strength, cooking loss, lightness, redness, and the preference value for texture and taste were influenced by the proportions of κ-carrageenan and the concentration of the pluchea leaf tea extract, respectively. The spider web graph showed that the K0L2 treatment had the largest area at 79.16 cm^2^ and a preference score of 15.8 (which can be assigned to the very like category), suggesting that this is the best treatment; this is in accordance with the results of the physicochemical and sensory tests, but it did not correlate with the highest bioactive content (TPC and TFC) and DPPH.

## Figures and Tables

**Figure 1 molecules-27-05062-f001:**
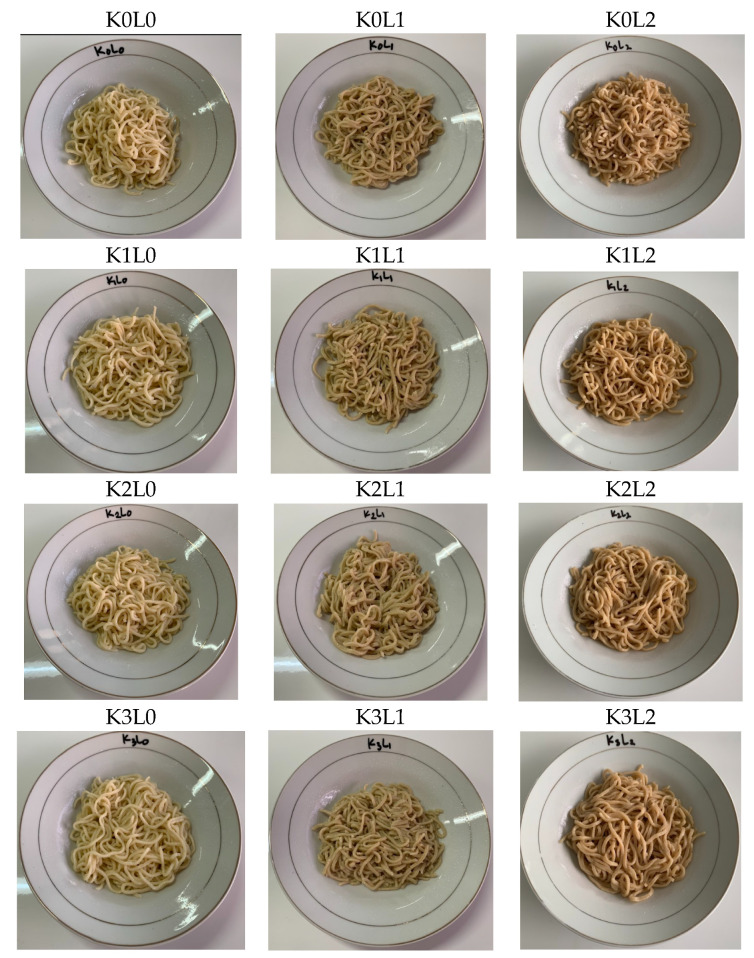
Stink lily noodles. Note that K0 = wheat flour: stink lily flour: κ-carrageenan = 80:20:0, K1 = wheat flour: stink lily flour: κ-carrageenan = 80:19:1, K2 = wheat flour: stink lily flour: κ-carrageenan = 80:18:2, K3 = wheat flour: stink lily flour: κ-carrageenan = 80:17:3, L0 = concentration of the hot water extract from the pluchea leaf tea = 0%, L1 = concentration of the hot water extract from the pluchea leaf tea = 15%, L2 = concentration of the hot water extract from the pluchea leaf tea = 30%.

**Figure 2 molecules-27-05062-f002:**
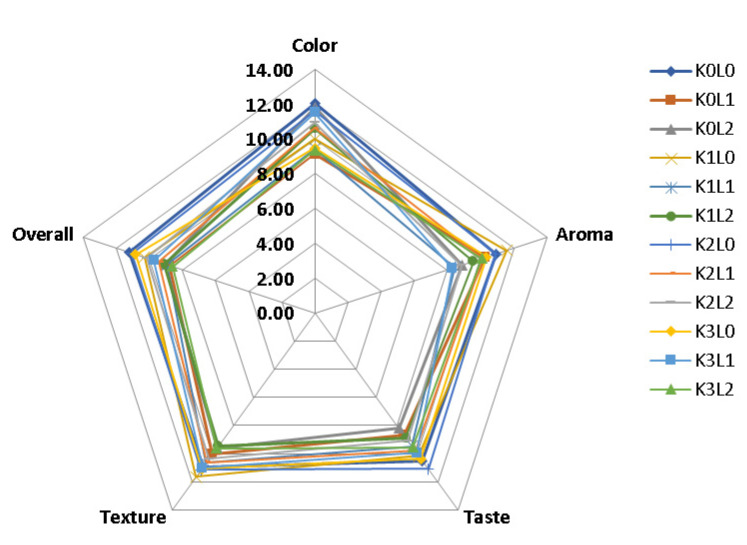
Spider web graph to determine the best treatment of the stink lily noodles.

**Table 1 molecules-27-05062-t001:** The formula of the hot water extract of the pluchea leaf tea.

Materials	Concentration of Hot Water Extract of Pluchea Leaf Tea (% *b*/*v*)		
	0	15	30
Pluchea leaf tea (g)	0	4.5	9
Hot water (mL)	30	30	30

**Table 2 molecules-27-05062-t002:** The formula of the stink lily wet noodles.

Material	K0L0	K0L1	K0L2	K1L0	K1L1	K1L2	K2L0	K2L1	K2L2	K3L0	K3L1	K3L2
Wheat flour (g)	120	120	120	120	120	120	120	120	120	120	120	120
Stink lily flour (g)	30	30	30	28.5	28.5	28.5	27	27	27	25.5	25.5	25.5
κ-Carrageenan (g)	0	0	0	1.5	1.5	1.5	3	3	3	4.5	4.5	4.5
Egg (g)	30	30	30	30	30	30	30	30	30	30	30	30
Salt (g)	3	3	3	3	3	3	3	3	3	3	3	3
Baking powder (g)	1.5	1.5	1.5	1.5	1.5	1.5	1.5	1.5	1.5	1.5	1.5	1.5
Water (mL)	30	0	0	30	0	0	30	0	0	30	0	0
Hot water extract of the pluchea leaves 15% (mL)	0	30	0	0	30	0	0	30	0	0	30	0
Hot water extract of the pluchea leaves 30% (mL)	0	0	30	0	0	30	0	0	30	0	0	30
Total (g)	214.5	214.5	214.5	214.5	214.5	214.5	214.5	214.5	214.5	214.5	214.5	214.5

Note: K0 = wheat flour: stink lily flour: κ-carrageenan = 80:20:0. K1 = wheat flour: stink lily flour: κ-carrageenan = 80:19:1. K2 = wheat flour: stink lily flour: κ-carrageenan = 80:18:2. K3 = wheat flour: stink lily flour: κ-carrageenan = 80:17:3. L0 = concentration of the hot water extract from the pluchea leaf tea = 0%. L1 = concentration of the hot water extract from the pluchea leaf tea = 15%. L2 = concentration of the hot water extract from the pluchea leaf tea = 30%.

**Table 3 molecules-27-05062-t003:** Color, moisture content, swelling index, cooking loss, and tensile strength of the stink lily noodles.

Samples	Color					Moisture Content (%wb)	Swelling Index (%)	Cooking Loss (%)	Tensile Strength (N)
	L*	a*	b*	C	h				
K0L0	73.00 ± 0.06	1.20 ± 0.06	16.90 ± 0.25 ^a^	17.00 ± 0.31 ^a^	86.40 ± 0.00 ^g^	64.15 ± 0.70	148.90 ± 0.15 ^c^	18.83 ± 0.44	0.106 ± 0.002
K0L1	68.70 ± 0.35	2.60 ± 0.06	26.50 ± 0.32 ^d^	26.50 ± 0.29 ^c^	84.40 ± 0.21 ^bc^	63.66 ± 0.38	146.36 ± 0.27 ^b^	19.06 ± 0.43	0.105 ± 0.001
K0L2	67.80 ± 0.20	2.80 ± 0.06	27.80 ± 0.46 ^ef^	27.80 ± 0.45 ^e^	84.20 ± 0.32 ^abc^	62.83 ± 0.58	142.25 ± 0.39 ^a^	20.13 ± 0.71	0.116 ± 0.006
K1L0	73.40 ± 0.25	1.30 ± 0.06	17.30 ± 0.15 ^ab^	17.30 ± 0.15 ^a^	85.70 ± 0.21 ^f^	64.42 ± 0.80	149.63 ± 0.34 ^d^	18.47 ± 0.31	0.086 ± 0.005
K1L1	69.00 ± 0.36	2.80 ± 0.12	27.30 ± 0.45 ^e^	27.40 ± 0.50 ^d^	84.20 ± 0.15 ^a^	62.95 ± 0.68	146.65 ± 0.43 ^b^	19.36 ± 0.92	0.103 ± 0.004
K1L2	68.30 ± 0.15	3.00 ± 0.12	28.60 ± 0.12 ^g^	28.70 ± 0.15 ^f^	84.00 ± 0.15 ^abc^	63.37 ± 1.04	148.85 ± 0.57 ^c^	19.76 ± 0.90	0.108 ± 0.005
K2L0	73.70 ± 0.10	1.50 ± 0.00	17.70 ± 0.26 ^bc^	17.80 ± 0.23 ^a^	85.20 ± 0.06 ^de^	64.67 ± 1.08	155.67 ± 0.46 ^h^	18.18 ± 0.45	0.098 ± 0.002
K2L1	69.40± 0.15	3.00 ± 0.06	28.20 ± 0.15 ^fg^	28.40 ± 0.15 ^f^	83.80 ± 0.17 ^ab^	63.74 ± 0.75	150.96 ± 0.71 ^e^	18.62 ± 0.41	0.106 ± 0.005
K2L2	68.70 ± 0.06	3.10 ± 0.15	29.30 ± 0.00 ^h^	29.40 ± 0.06 ^g^	84.00 ± 0.15 ^abc^	64.25 ± 1.60	154.82 ± 0.44 ^g^	19.34 ± 0.77	0.114 ± 0.003
K3L0	74.50 ± 0.23	1.70 ± 0.12	18.10 ± 0.00 ^c^	18.10 ± 0.06 ^b^	84.60 ± 0.21 ^cd^	65.83 ± 0.22	162.21 ± 0.25 ^i^	17.83 ± 0.41	0.110 ± 0.003
K3L1	69.80 ± 0.50	3.20 ± 0.06	29.30 ± 0.12 ^h^	29.30 ± 0.06 ^g^	83.80 ± 0.53 ^ab^	64.57 ± 1.78	153.35 ± 0.15 ^f^	18.36 ± 0.17	0.124 ± 0.007
K3L2	69.00 ± 0.20	3.30 ± 0.26	30.00 ± 0.06 ^i^	30.10 ± 0.06 ^h^	83.60 ± 0.06 ^a^	65.49 ± 1.04	159.59 ± 0.52 ^i^	19.22 ± 0.84	0.126 ± 0.008

* The results were presented as SD of the means that were achieved by quadruplicate. Means with different superscripts (alphabets) in the same column are significantly different, *p* < 5%.

**Table 4 molecules-27-05062-t004:** Total phenol content, total flavonoid content, and the DPPH free radical scavenging activity of the stink lily noodles.

Samples	TPC (mg GAE/g Dry Noodles)	TFC (mg CE/g Dry Noodles)	DPPH Scavenging Activity (mg GAE/g Dry Noodles)
K0L0	0.341 ± 0.034 ^a^	0.057 ± 0.004 ^a^	0.414 ± 0.006 ^a^
K0L1	0.948 ± 0.027 ^d^	0.704 ± 0.007 ^c^	0.742 ± 0.016 ^c^
K0L2	1.099 ± 0.047 ^e^	1.109 ± 0.007 ^d^	0.757 ± 0.001 ^c^
K1L0	0.458 ± 0.040 ^b^	0.082 ± 0.002 ^a^	0.523 ± 0.026 ^b^
K1L1	0.924 ± 0.077 ^d^	0.735 ± 0.003 ^c^	0.750 ± 0.006 ^c^
K1L2	1.005 ± 0.070 ^d^	1.091 ± 0.095 ^d^	0.751 ± 0.001 ^c^
K2L0	0.500 ± 0.045 ^b^	0.089 ± 0.008 ^a^	0.532 ± 0.005 ^b^
K2L1	0.965 ± 0.025 ^d^	0.718 ± 0.018 ^c^	0.748 ± 0.004 ^c^
K2L2	1.196 ± 0.027 ^f^	1.336 ± 0.046 ^e^	0.758 ± 0.009 ^c^
K3L0	0.527 ± 0.007 ^b^	0.218 ± 0.003 ^b^	0.587 ± 0.020 ^b^
K3L1	0.807 ± 0.031 ^c^	0.726 ± 0.039 ^c^	0.751 ± 0.008 ^c^
K3L2	1.145 ± 0.064 ^ef^	1.067 ± 0.063 ^d^	0.751 ± 0.007 ^c^

Note: The results were presented as SD of the means that were achieved by quadruplicate. Means with different superscripts (alphabets) in the same column are significantly different, *p* < 5%.

**Table 5 molecules-27-05062-t005:** Sensory properties of the stink lily noodles.

Samples	Hedonic Preference Score				
	Color	Aroma	Taste	Texture	Overall Acceptance
K0L0	12.02 ^f^	10.93 ^d^	10.52	10.95	11.24 ^f^
K0L1	9.12 ^a^	10.26 ^cd^	8.72	10.03	8.79 ^a^
K0L2	11.83 ^ef^	8.86 ^ab^	8.18	9.58	9.04 ^ab^
K1L0	9.98 ^bc^	11.58 ^e^	10.15	11.66	10.3 ^cde^
K1L1	9.44 ^ab^	9.58 ^bc^	9.49	10.66	8.93 ^ab^
K1L2	10.59 ^cd^	8.45 ^a^	8.87	9.46	9.11 ^ab^
K2L0	11.62 ^f^	10.93 ^de^	11.08	11.15	11.07 ^def^
K2L1	10.63 ^cd^	10.14 ^ab^	9.82	10.61	9.42 ^abc^
K2L2	10.94 ^de^	8.84 ^cd^	9.09	10.38	10.16 ^cde^
K3L0	11.51 ^ef^	10.36 ^cd^	10.37	11.07	10.84 ^def^
K3L1	9.37 ^ab^	10.07 ^a^	9.95	11	9.72 ^bc^
K3L2	9.47 ^ab^	8.29 ^cd^	9.57	9.64	8.62 ^a^

Note: The results were presented as SD of the means that were achieved by quadruplicate. Means with different superscripts (alphabets) in the same column are significantly different, *p* < 5%.

## Data Availability

Data reported in this study are contained within the article. The underlying raw data are available on request from the corresponding author.

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
