# Peer review of "The Effect of κ-Carrageenan Proportion and Hot Water Extract of the Pluchea indica Less Leaf Tea on the Quality and Sensory Properties of Stink Lily (Amorphophallus muelleri) Wet Noodles"

_molecules, 2022, doi:10.3390/molecules27165062_

Round 1

Reviewer 1 Report

The study has some interesting information. However, it needs to be clear in the introduction why k-carrageenan and the extract are added to the noodles, what defects of the noodles are to be improved, and what functions the addition of the extract makes edible, not simply the determination of the antioxidant properties in the noodles.

1.    L 205-206 revise the sentence.

2.      3. Is there a standard for the maximum allowable amount of k-carrageenan in noodles?

3.      4.1. Cooking Quality provides an extensive discussion of the effects of k- carrageenan, including its use in other studies, but fails to provide a focused account of what was seen in the different samples in this study, and it is recommended that an analysis specific to the results of this study be included in the discussion.

4.      Bioactive Compounds and Antioxidant Activity

What is the significance of this section on TPC, TFC and DPPH of stink lily wet noodles? Is there any loss of activity after boiling? Is it still meaningful to eat after the loss?

Author Response

Dear Reviewer 

Thank you for your suggestions to improve our manuscript. Attached I send the manuscript that I have revised. The sections I am revising are marked in yellow in the manuscript.
There are several explanations regarding reviewer questions
1. The maximum amount of k-carrageenan allowed is not clear, but the concentration of k-carrageenan for noodle products is a maximum of 3%
2. Testing of the bioactive compound content and antioxidant activity (DPPH) was only carried out on cooked noodles, while raw wet noodles were not carried out, so it is not known the number loss of bioactive compounds and antioxidant activity.

The Best Regards

Paini Sri Widyawati

Reviewer 2 Report

I think that the topic is very interesting for the journal fits perfectly in the Special Issue to which it has been submitted, since it provides the possibility of increase the functional value of wet noodle products.

However, there are important aspects that must be clarified to be published:

- LINE 3. Replace "to" with "on".

- LINES 13 and 14. What do you mean with "b/b" and "b/v"? Review it throughout the text.

- LINE 15. Put Anova in capital letters.

- LINE 21. The DPPH free radical scavenging assay is usually associated with the acronym DPPH and not with AOA.

- LINE 76. The figures included in the non-published material could be included in the manuscript to better understand the elaboration process.

- LINE 101. The information included in the manuscript seems misplaced. Introduce tables and figures as close as possible to where they are cited in the text.

- LINE 121. Remove repeated words.

- LINE 141. Include the name of the authors, replacing “by [27]” with “by Rathorel et al. [27]”. Review it throughout the text.

- LINE 173-185. The experimental design should be explained in more detail: Detail hedonic scale scoring method.

- LINES 194-197. Explain better the statistical analysis. Were you do a pre-treatment of the data (normal distribution and variance homogeneity)? Were the values considered significant when P = 0.05 or when P < 0.05?

- LINE 238. Figure 1b is not clear. Please replace it with a higher quality one. Moreover, reflect in the legend that there is Figure 1a and 1b.

- LINE 243. Please, use superscripts in the table. The same in Table 5.

- LINE 289. Revise the number of significant decimals.

Author Response

Dear Reviewer 

Thank you for your suggestions to improve our manuscript. Attached I send the manuscript that I have revised. The sections I am revising are marked in yellow in the manuscript.
We have changed, corrected, and added information according to the reviewer's suggestions

The Best Regards

Paini Sri Widyawati

Round 2

Reviewer 1 Report

The author has sufficiently revised the article for publication.

Author Response

Dear Reviewer 

Thanks for attention

The Best Regards

Paini Sri Widyawati

Reviewer 2 Report

Although the manuscript has been considerably improved, the authors have not taken into account all the suggestions made by the reviewer. Moreover, it continues to have quite a few formatting errors. Therefore, there are still important aspects that must be clarified to be published:

- LINE 76. The figures included in the previous non-published material version could be included in the manuscript to better understand the elaboration process. Include a flowchart of the Stink Lily (Amorphophallus Muelleri) wet noodles making process.

- LINE 107. The information included in the manuscript seems misplaced. Put tables and figures as close as possible to where they are cited in the text. For example, Table 2 is cited on line 107, but does not appear until line 244.

- LINE 304. Revise the number of significant decimals in Table 4.

- LINE 350. Include the name of the authors, replacing “[13]” with “by Li et al. [13]”. Review it throughout the text (LINES 355, 356, 358, 360, 362, 373, 394, 408, 434, 445, 446, 448, 451, 476, 477, 481, 483).

- LINE 385. Begin the sentence as “k-carrageenan can trap the free water molecule…”, and move [13,46,41] to the end of the sentence.

- LINE 388. Include “other authors” after “Whereas”.

- LINE 408. Include “other authors” after “by”. The same in LINE 417, 439.

- LINE 435. Include “other authors” before “[16, 17,18,19]”. The same in LINE 464, 501.

- LINE 489. Include “other authors” after “to”.

Author Response

Dear Reviewer

I have corrected the manuscript that I submitted to MDPI, in the parts that I revised I put a yellow mark.
The complete manuscript that I have corrected is attached

Thanks for attention

The Best Regards

Paini SW
